# Small molecule-mediated refolding and activation of myosin motor function

Michael B Radke[1†], Manuel H Taft[1†], Britta Stapel[2], Denise Hilfiker-Kleiner[2], Matthias Preller[1,3], Dietmar J Manstein[1,4]*

[1]Institute for Biophysical Chemistry, Hannover Medical School, Hannover, Germany; [2]Department of Cardiology and Angiology, Hannover Medical School, Hannover, Germany; [3]Centre for Structural Systems Biology, German Electron Synchrotron (DESY), Hamburg, Germany; [4]Research Division for Structure Analysis, Hannover Medical School, Hannover, Germany

**Abstract** The small molecule EMD 57033 has been shown to stimulate the actomyosin ATPase activity and contractility of myofilaments. Here, we show that EMD 57033 binds to an allosteric pocket in the myosin motor domain. EMD 57033-binding protects myosin against heat stress and thermal denaturation. In the presence of EMD 57033, ATP hydrolysis, coupling between actin and nucleotide binding sites, and actin affinity in the presence of ATP are increased more than 10-fold. Addition of EMD 57033 to heat-inactivated β-cardiac myosin is followed by refolding and reactivation of ATPase and motile activities. In heat-stressed cardiomyocytes expression of the stress-marker atrial natriuretic peptide is suppressed by EMD 57033. Thus, EMD 57033 displays a much wider spectrum of activities than those previously associated with small, drug-like compounds. Allosteric effectors that mediate refolding and enhance enzymatic function have the potential to improve the treatment of heart failure, myopathies, and protein misfolding diseases.

**\*For correspondence:** manstein. dietmar@mh-hannover.de

†These authors contributed equally to this work

**Competing interests:** The authors declare that no competing interests exist.

**Reviewing editor**: Jeffrey Kelly, Scripps Research Institute, United States

## Introduction

Mammalian cells typically produce in excess of 10,000 different proteins displaying enzymatic activities. To retain the conformational flexibility required for their catalytic function, most of these enzymes are only marginally thermodynamically stable. Their 'native state' corresponds to an ensemble of conformers sharing closely related three-dimensional structures (*Bartlett and Radford, 2009*; *Hartl et al., 2011*). External stress brought about by changes in pH, temperature, ionic strength, and the composition and concentration of small-molecule ligands affects the transition to the denatured state. The denatured state is characterized by the loss of enzymatic activity and can be described as an ensemble of states with high conformational entropy (*Anfinsen and Scheraga, 1975*). Denatured enzyme molecules are more likely to be affected by adsorption, aggregation, precipitation, and proteolysis. At the cellular level, their aggregation with one another or with properly functioning proteins leads to deleterious consequences. Their accumulation can interfere with proteasome function, disrupt normal cellular processes by binding critical cell-signaling and cell-trafficking molecules, and trigger signals that result in cell death (*Tyedmers et al., 2010*; *Willis and Patterson, 2013*). The resulting proteotoxicity has been implicated in the pathogenesis of proteinopathies such as sickle cell anemia, cystic fibrosis, various forms of amyloidosis, and Alzheimer's and Parkinson's disease. Elaborate cellular systems for the maintenance of protein homeostasis (proteostasis) have evolved to prevent the accumulation of unfolded or misfolded protein aggregates. These comprise approximately 200 general and specialized chaperone components and 600 ubiquitin-proteasome system (UPS) and autophagy system components that support protein folding, refolding of stress-denatured enzymes, disaggregation, and proteolytic degradation of irreversibly misfolded proteins (*Nalepa et al., 2006*; *Tyedmers et al., 2010*).

**eLife digest** Our muscles contain large numbers of 'motor proteins' called myosins. To contract a muscle, many myosin molecules expend energy to 'walk' along a filament made from another molecule, called actin, and generate a pulling force. Like other proteins, myosins must fold into the correct shape to work, but high temperatures or other types of stress can disrupt their ability to adopt or maintain the correct shape. Misfolding of myosins, for example, can result in muscular diseases, including those that affect the heart; so there is an ongoing effort to find compounds that can stabilize protein folding and treat these diseases.

The small molecule EMD 57033 was discovered over 20 years ago, and its ability to increase the strength of muscle contractions suggested that it could be used to treat chronic heart failure, but the risk of side effects limited its clinical use. The effectiveness of other compounds that improve cardiac muscle function is still routinely compared to EMD 57033, however the exact mechanism responsible for its effect on muscle tissue remained unknown.

Now Radke, Taft et al. have identified the part of the myosin protein that EMD 57033 binds to, and shown how this activates muscle contraction. The experiments also, unexpectedly, revealed that EMD 57033 is able to convert misfolded myosin back into the fully functional form. By revealing this refolding effect, the findings of Radtke, Taft et al. suggest that similar small molecules could be used as drugs for the treatment of protein misfolding diseases, muscular diseases, and heart failure.

Myosins are amongst the most abundant proteins in our body. In addition to their role as molecular motors driving the stiffening and contraction of muscles, they contribute to a wide range of functions. In the case of myosin, the negative consequences of misfolding are amplified by the formation of roadblock-like, strongly-bound complexes with their actin filament tracks. Aberrant myosin activity and proteostasis contribute to hereditary skeletal myopathies, cardiomyopathies, and various disorders of the nervous system (*Oldfors, 2007*; *Walsh et al., 2010*; *Getty and Pearce, 2011*; *Getty et al., 2011*). In the course of investigations to identify small-molecule ligands that stabilize myosin against thermal denaturation, we identified the thiadiazinone derivative EMD 57033 as a potent mediator of myosin folding. EMD 57033 is the positive optical isomer of the racemate EMD 53998. The negative isomer, EMD 57439, acts as a potent phosphodiesterase III inhibitor (*Gambassi et al., 1993*). EMD 57033 displays only minimal phosphodiesterase III inhibitory effects and has been described to act both as $Ca^{2+}$ sensitizer and activator of the actomyosin interaction (*Beier et al., 1991*; *Ferroni et al., 1991*; *Gambassi et al., 1993*; *Solaro et al., 1993*). Studies with EMD 57033 performed in dogs, with heart failure induced through long-term tachycardia pacing, show potent positive inotropic effects, reduced oxygen cost of contraction, and enhanced diastolic chamber performance (*Senzaki et al., 2000*). It has been postulated that the observed $Ca^{2+}$ sensitization is closely linked to EMD 57033-induced changes in myosin cross-bridge kinetics that interfere with normal troponin–tropomyosin regulation (*Solaro et al., 1993*). Here, we provide evidence indicating that EMD 57033 binds to an allosteric pocket in the myosin motor domain near the base of the lever arm. In addition to increasing motility, force production, and thermal stability of native myosin, binding of EMD 57033 to heat-inactivated myosin induces refolding and reactivation of ATPase activity and motility. Our findings demonstrate that small molecules such as EMD 57033 can stabilize, enhance the activity of and correct stress-associated misfolding of globular enzymes. Pharmacological chaperones with the activity profile of EMD 57033 open the door to a new class of therapeutic drugs and the production of commercially viable biosensors, as described in the 'Discussion'.

## EMD 57033 acts as an allosteric activator of the myosin motor

We used microscale thermophoresis to show that EMD 57033 binds purified β-cardiac myosin S1 with a stoichiometry of one molecule of EMD 57033 per myosin head with an affinity of 7.3 ± 1.9 µM (*Figure 1A*). Direct binding of EMD 57033 to the motor domain was observed for *Dictyostelium discoideum* (*Dd*) myosin-2 and myosin-5 motor domain constructs but not for *Dd* myosin-1B, *Dd* myosin-1C, *Dd* myosin-1D, and *Dd* myosin-1E motor domain constructs. Myosin constructs that bind EMD 57033 display increased basal and actin-activated ATPase activity in the presence of the compound. The most potent interaction was observed with β-cardiac myosin with half-maximal activation ($AC_{50}$) observed at 7.0 ± 1.5 µM EMD 57033 (*Table 1*). The activation of skeletal muscle myosin-2 and *Dd* myosin-5 is

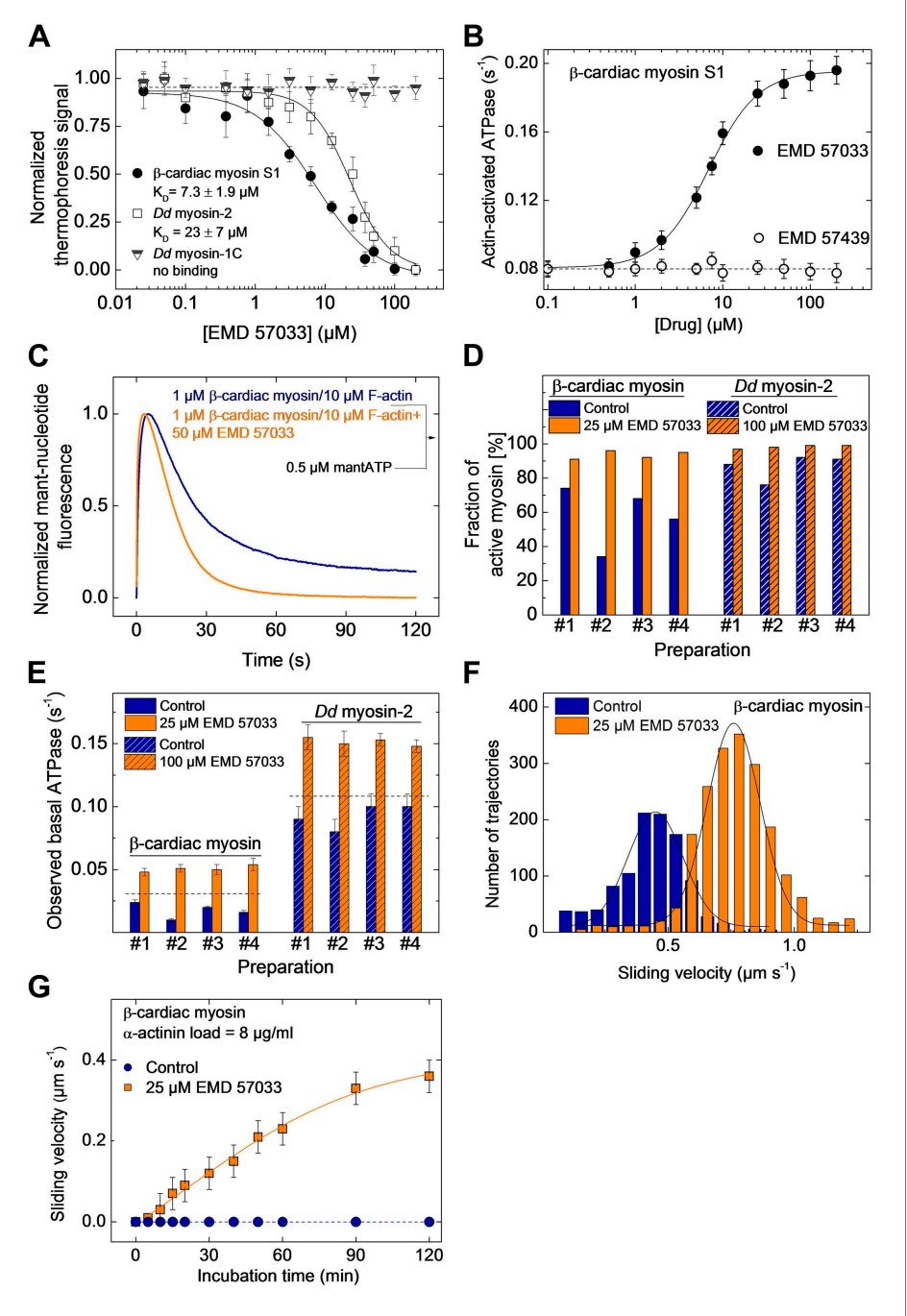

**Figure 1**. EMD 57033 binds to the myosin motor domain and activates ATPase and motor activities. (**A**) Direct interaction study between fluorescently labeled β-cardiac myosin S1, *Dd* myosin-2 motor domain, and *Dd* myosin-1C motor domain and EMD 57033. The myosin concentration was kept constant at 100 nM and EMD 57033 was titrated from 10 nM to 200 μM. The normalized thermophoresis signals were plotted against the EMD 57033 concentration. $K_D$ values were obtained by fitting the data to the Hill equation. Error bars indicate SD (n = 3). (**B**) Dose-dependent activation of the actin-activated ATPase of β-cardiac myosin S1 by EMD 57033. Control measurements with the (−)-enantiomer EMD 57439 show no activation. Errors indicate s.d. (n = 4). (**C**) Single-turnover analysis of mantATP binding, hydrolysis, and product release by β-cardiac myosin in the absence (blue curve) and presence of 25 μM EMD 57033 (orange curve). (**D**) Fraction of active myosin heads in the absence and presence of saturating concentrations of EMD 57033 determined by active site titrations. Four β-cardiac myosin and four *Dd* myosin-2 preparations were compared. In the absence of EMD 57033 (blue bars) β-cardiac

*Figure 1. Continued on next page*

*Figure 1. Continued*

myosin preparations display between 34% and 74% and *Dd* myosin-2 preparations 76–92% activity. The fraction of active protein increased to 87–100% following the addition of EMD 57033 (25 µM for β-cardiac myosin and 100 µM for *Dd* myosin-2). Addition of EMD 57033 to fully denatured myosin aggregates followed by incubation for 6 hr on ice or at 20°C produced no detectable recovery of enzymatic activity. (**E**) Activation of basal ATPase activity by EMD 57033. The observed turnover of ATP is 1.4-fold (*Dd* myosin-2) and 1.5-fold (β-cardiac myosin) higher than expected for preparations of 100% active protein (dashed line). (**F**) EMD 57033-mediated activation of motor activity. The histograms and Gaussian fits show the distribution and average sliding velocity of actin filaments on lawns of β-cardiac myosin in the absence (blue histograms) and presence of EMD 57033 (orange histograms). Errors indicate SD (**G**) EMD 57033-mediated increase in force production. A constant frictional load of 8 µg/ml α-actinin was applied to stall β-cardiac myosin-based motility in the absence of EMD 57033. Filament movement is restarted after the addition of EMD 57033. Errors indicate SD.

The following figure supplements are available for figure 1:

**Figure supplement 1**. EMD 57033-mediated activation of motor activity for a recombinant *Dd* myosin-2 motor domain construct.

**Figure supplement 2**. Frictional loading experiments.

twofold and fivefold weaker. Neither binding nor activation was observed for the (−)-enantiomer EMD 57439 (*Figure 1B*). The consequences of EMD 57033 binding on the turnover of ATP by actin-activated β-cardiac myosin were followed in single-turnover experiments. Using the extrinsic fluorescence probe 2′-/3′-O-(N′-methylanthraniloyl)-ATP (mantATP) instead of ATP as substrate, we observed signal changes that can be attributed to three phases. The initial fast rise of fluorescence reflects the binding of the ATP analogue to the myosin active site, the following plateau phase monitors the duration of the hydrolysis reaction, and the third phase, corresponding to a decrease in fluorescence signal intensity, monitors the rate of product release. In the presence of EMD 57033, we observed an approximately twofold net increase in the rate of ATP turnover (*Figure 1C*).

The number of experimentally accessible active sites is always smaller in preparations of purified enzymes than the number of active sites calculated based on protein concentration. The observed deviation results from incomplete folding, the presence of impurities, and the stress-induced partial loss of function during purification and storage. In the case of myosin, the amplitude of the fast rise in fluorescence intensity that follows binding of mantATP can be used to estimate the fraction of functional protein by active site titration (*Tsiavaliaris et al., 2002*). *Figure 1D* shows the results of active site titrations for four typical preparations of β-cardiac myosin and a *Dd* myosin-2 motor domain construct. Dependent on the type of myosin, purification conditions, and length of storage, the fraction of active myosin heads varies between 32% and 92%. The fraction of active sites increased to 87–98% for the same enzyme preparations following the addition of EMD 57033. The increase in the number of binding-competent active sites is followed by a correlated increase in myosin ATPase activity (*Figure 1D,E*). In addition to the apparent conversion of inactive to catalytically competent myosin, ATPase activities measured in the presence of EMD 57033 correlate with the total number of myosin heads rather than

**Table 1.** Interaction of EMD 57033 with myosin isoforms

| Myosin construct | AC$_{50}$ ATPase | Normalized maximal ATPase activation (basal) | Normalized maximal ATPase activation (with 30 µM actin) | Binding affinity (MST) |
|---|---|---|---|---|
| β-Cardiac myosin-2 (S1, full-length) | 7.0 µM | 1.5 | 2.5 | 7.3 µM |
| Skeletal muscle myosin-2 (HMM) | 15.1 µM | 1.6 | 2.2 | – |
| *Dd* myosin-2 motor domain | 25.8 µM | 1.4 | 2.8 | 23.0 µM |
| *Dd* myosin-5b motor domain | 35.4 µM | 1.4 | 1.6 | – |
| *Dd* myosin-1B, -1C, -1D, -1E motor domains | n.a. | no effect | | no binding (myosin-1C/-1D/-1E) |
| *Dd* myosin-2 ΔSH3 [22] (lacks residues 33–79) | n.a. | no effect | | n.a. |

the initial number of active myosin heads. Moreover, the measured ATPase activities exceed the values expected for a 100% active population of myosin heads by 30–50% (*Figure 1D*).

To probe the effect of EMD 57033 on myosin motor activity, we performed in vitro motility assays. The sliding velocity of actin filaments increased 1.8-fold for β-cardiac myosin (*Figure 1F*) and 1.6-fold for a *Dd* myosin-2 motor domain construct (*Figure 1—figure supplement 1*). In addition, the fraction of moving filaments increased from <75% to >95%. To examine whether EMD 57033 affects force production by β-cardiac myosin as well, we performed frictional loading experiments (*Greenberg and Moore, 2010*). In initial experiments, we determined the minimal concentration of α-actinin molecules that generate sufficient load to stall the movement of actin filaments for a given surface density of myosin heads. In the presence of EMD 57033, the α-actinin concentration required to stall filament movement increased from 8 µg/ml to ≥12 µg/ml (*Figure 1—figure supplement 2*). Next, we generated in vitro motility flow cells with actin filaments held at stall force. The addition of 25 µM EMD 57033 to these flow cells restarts filament movement. The time-dependence of the recovery of motile activity is best fit by a hyperbola. A plateau value of 0.4 µm s$^{-1}$ is reached after more than 2 hr (*Figure 1G*).

## Localization of the EMD 57033 binding pocket

Steady-state kinetic assays performed in the absence and the presence of EMD 57033 indicate an allosteric mode of action (*Figure 2—figure supplement 1*). The binding site of EMD 57033 in the human β-cardiac myosin motor domain was further defined by molecular docking experiments in combination with direct binding studies using myosin motor domain constructs with alterations in the predicted binding region. Initial blind docking to homology models of the β-cardiac myosin motor domain in the post-rigor state indicated preferred binding of the drug to a region near the small N-terminal, SH3-like βbarrel subdomain. Important contacts are predicted to involve the SH3-like subdomain (residues 34–72) and two α-helices (residues 20–28 and 98–111). The mode of binding and the orientation of EMD 57033 in this pocket were further analyzed using a flexible, targeted docking procedure. Rearrangements in the side chains of residues Arg29 and Lys34 are predicted to bring these two amino acids into the close vicinity of polar groups in EMD 57033. The identified binding pocket exhibits a Y shape, supporting two slightly different clusters of binding poses of EMD 57033 with predicted binding free energies in the range of −11 to −12 kcal mol$^{-1}$. The sets of binding poses differ mainly in the orientation of the dimethylated catechol group, occupying either of the two possible branches of the Y, while the thiadiazinone moiety appears little affected and resides in the root of the Y shaped cavity (*Figure 2A,B*). Potential residues involved in the binding of the thiadiazinone and tetrahydroquinoline moieties of the compound include Arg23, Asp85, Lys86, Asp107, Arg108, and Ser111. The dimethylated catechol moiety is mainly bound by residues Lys34, Lys48, and Gln79, and thus, interacts with the SH3-like β-barrel. The contribution of residues from the SH3-like β-barrel to EMD 57033 binding is in good agreement with our observation that all class-1 myosins tested, which lack this subdomain, fail to bind EMD 57033 (*Table 1*). To verify that the SH3-like β-barrel is necessary to mediate EMD 57033 binding, we performed direct binding, ATPase, and in vitro motility assays with *Dd* myosin-2 ΔSH3, a recombinant motor domain construct that lacks this subdomain (*Fujita-Becker et al., 2006*). We observed neither binding of EMD 57033 to *Dd* myosin-2 ΔSH3 nor EMD 57033-induced activation of the constructs chemomechanical activity (*Figure 2C,D* and *Table 1*). Further studies using point mutations that eliminate key residues predicted to interact with EMD 57033 are needed to confirm the binding site.

## EMD 57033-mediated changes in the kinetic behavior of the myosin motor

To gain a better understanding of its mechanism of action, we examined the effect of EMD 57033-binding on the kinetic activity of the myosin motor. The effect of EMD 57033 on individual steps of the ATPase cycle of β-cardiac myosin was analyzed with the help of transient kinetic measurements and evaluated according to the model shown in *Figure 3A*, Scheme 1. Upon addition of 25 µM EMD 57033 to β-cardiac myosin, we observed a 1.6-fold increase in k$_{cat}$, the maximum rate of ATP turnover, and a 10-fold decrease in K$_{M(actin)}$, the F-actin concentration necessary for half-maximal activation (*Figure 3B*). The extent to which coupling between the actin and nucleotide binding sites is improved in the presence of EMD 57033 is indicated by the resulting 16-fold increase in the apparent second-order rate constant (k$_{cat}$/K$_{M(actin)}$) for the actin-activated ATPase activity (*Table 2*). Fast mixing of excess ATP with β-cardiac myosin results in an increase of the protein's intrinsic tryptophan fluorescence. The kinetic model predicts the double exponential 'behavior' of the observed transients to be associated with two discrete processes (*Figure 3C*). The fast process corresponds to the binding of ATP to the active site. The rate of the slow

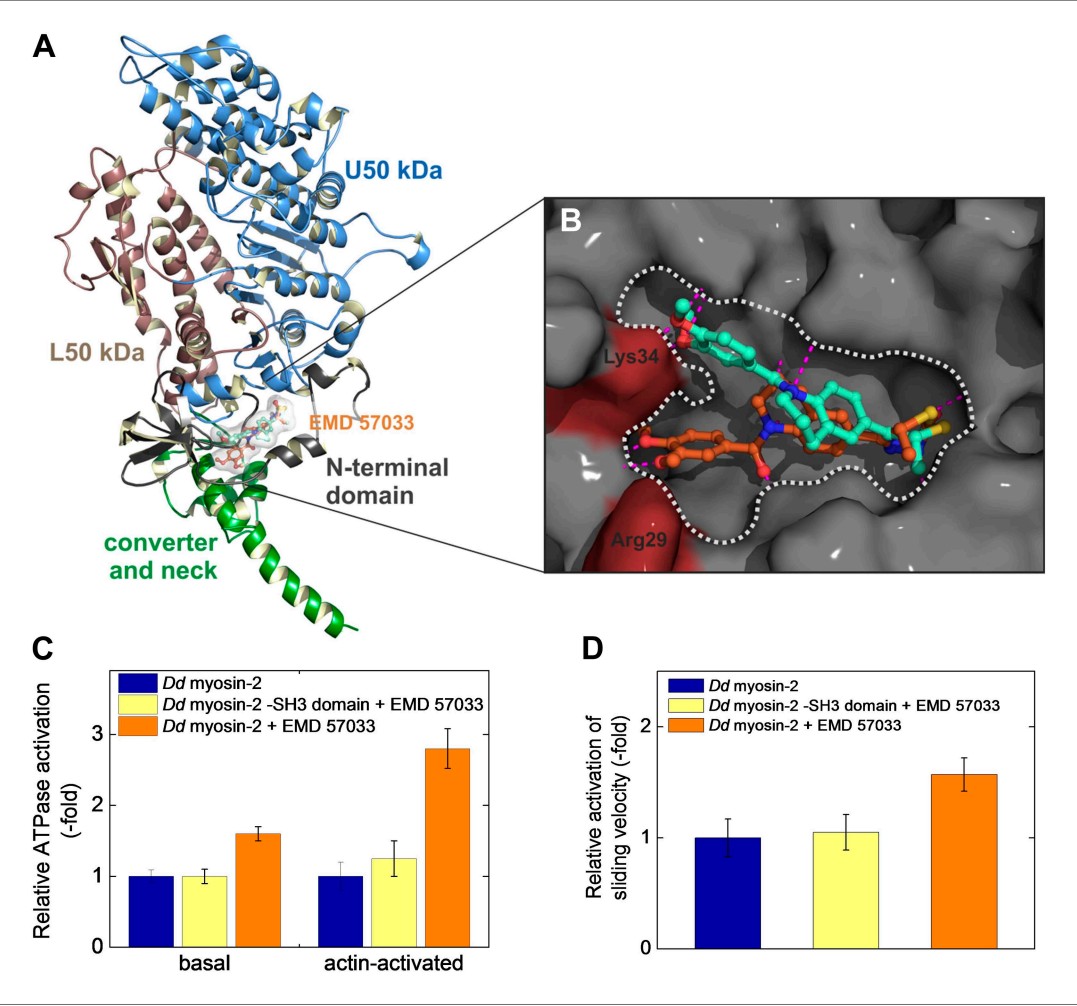

**Figure 2**. Localization of the EMD 57033 binding pocket. (**A**) Predicted binding of EMD 57033 near the base of the lever arm as obtained by molecular docking. The overview of the *Hs* β-cardiac myosin head domain includes residues 1–800. Two slightly different clusters of binding poses were identified for EMD 57033 with comparable binding affinities. (**B**) Close-up of the allosteric binding pocket shown in surface representation. The Y shape of the pocket is outlined and the two residues—Arg29 and Lys34—that were allowed full conformational flexibility during docking are colored red. For clarity only the best poses for the two identified clusters of binding modes are shown. Hydrogen bonds of EMD 57033 to the motor protein are shown in magenta. (**C**) The SH3-like subdomain is required for EMD 57033 binding and myosin activation. A truncated *Dd* myosin-2 motor domain construct without SH3-like β-barrel shows no significant activation in the presence of 100 µM EMD 57033. Errors indicate SD (n = 4). (**D**) In contrast to the wild-type *Dd* myosin-2 motor domain construct, no actin filament sliding velocity enhancement was observed upon addition of 100 µM EMD 57033 to the construct with truncated N-terminal SH3-like β-barrel. Errors indicate SD.

The following figure supplements are available for figure 2:

**Figure supplement 1**. Kinetic analysis of the effect of EMD 57033 binding using the Lineweaver–Burk plot.

process saturates at ATP concentrations above 100 µM and can be attributed to the ATP hydrolysis step (*Deacon et al., 2012*). Accordingly, EMD 57033 triggers a 3.4-fold increase in the second order rate constant for ATP binding ($K_1k_{+2}$), a 3.8-fold increase in the rate of the subsequent conformational change ($k_{+2}$), and a 12.9-fold increase in the rate of ATP hydrolysis ($k_{+3} + k_{-3}$) (*Figure 3D,E* and *Table 2*). Phosphate release is the rate-limiting step of the β-cardiac myosin actin-activated ATPase cycle. It can be monitored by the fluorescence increase that is associated with the much faster binding of phosphate to the A197C mutant of *Escherichia coli* phosphate binding protein, labeled at Cys197 with the thiol reactive coumarin dye MDCC (*Brune et al., 1994*) (*Figure 3F*). The observed transients in the absence and presence of

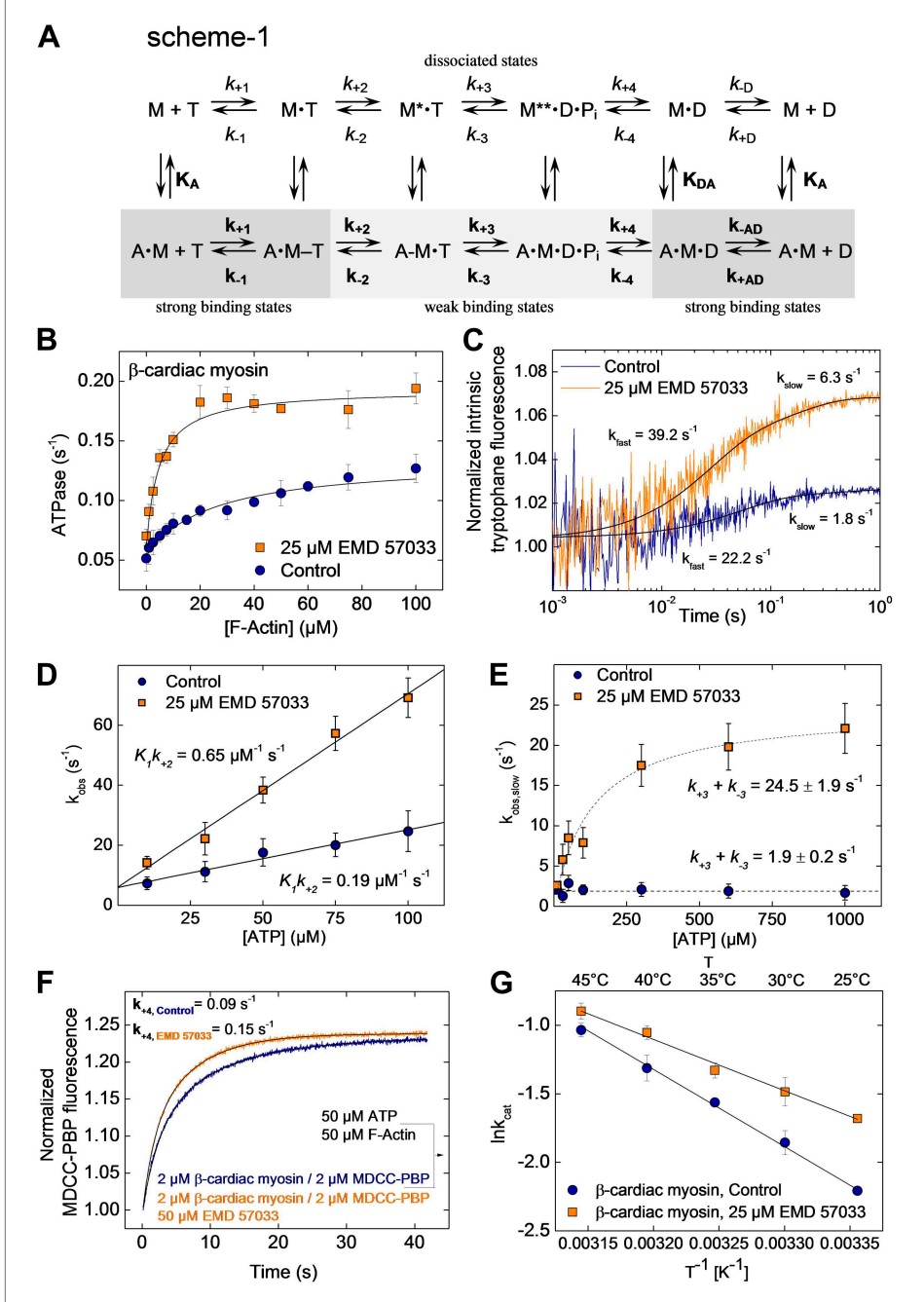

**Figure 3**. EMD 57033-mediated changes in the interaction with nucleotides and F-actin. (**A**) Kinetic reaction scheme of the actomyosin ATPase cycle. 'A' refers to actin, 'M' to myosin, 'T' to ATP, and 'D' refers to ADP. Rate constants are referred to as $k_{+n}$ and $k_{-n}$ and are assigned to the corresponding forward and reverse reactions. An additional notation is used that distinguishes between the constants in the absence and presence of actin by italic type ($k_{+1}$, $K_1$) and bold ($\mathbf{k_{+1}}$, $\mathbf{K_1}$), respectively; subscript A refers to actin ($\mathbf{K_A}$) and subscript D ($K_D$) refers to ADP (**B**) EMD 57033-mediated increase in the actin-dependence of ATP-turnover. In the presence of 25 µM EMD 57033, the plateau value representing $k_{cat}$ is 1.6-fold increased and the apparent actin affinity $K_{M(Actin)}$, indicated by the half-maximal activation of ATPase activity, is 10-fold increased (values are given in ***Table 2***). Errors indicate SD (n = 5). (**C**) EMD 57033-mediated increase in the rate of ATP binding and hydrolysis. The fast phase of the biphasic transients correspond to ATP binding, the slow phase can be attributed to ATP hydrolysis. (**D**) EMD 57033-mediated changes in the rate of ATP binding. The observed rates for the fast phase show a linear dependence on ATP concentration. Differences in the slopes indicate a threefold increase of the second order rate constants for ATP

*Figure 3. Continued on next page*

*Figure 3. Continued*

binding in the presence of EMD 57033. Errors indicate SD. (**E**) EMD 57033 mediates a 10-fold increase in the rate of ATP hydrolysis. The plateau values from a hyperbolic fit of the observed slow components of the fluorescence transients plotted against ATP concentrations define the rate of ATP hydrolysis. Errors indicate SD. (**F**) The rate of phosphate release increased 1.7-fold in the presence of EMD 57033. Phosphate release from β-cardiac myosin was observed following rapid mixing with excess ATP and F-actin in the presence of the fluorescent phosphate-binding protein MDCC-PBP. Errors indicate SD. (n = 6 from three different protein preparations). (**G**) Arrhenius analysis of the temperature dependence of β-cardiac myosin ATPase activity. The activation energy for the reaction is 1.5-fold reduced in the presence of EMD 57033. Errors indicate SD. (n = 3).

EMD 57033 were best fitted by single exponential functions. The resulting values for $k_{+4}$ indicate a 1.7-fold increase in the presence of the small compound. The subsequent ADP-release step ($k_{-D}$, $k_{-AD}$) appears unaffected by EMD 57033-binding (**Table 2**).

To assess the effect of EMD 57033 on the activation energy of ATP turnover by β-cardiac myosin, we recorded the temperature dependence of the reaction over the range 25–45°C. The resulting Arrhenius plot shows that EMD 57033 affects the slope of the recorded linear dependencies, indicating that the Arrhenius energy ($E_a$) is decreased from 47 kJ mol$^{-1}$ to 31 kJ mol$^{-1}$ in the presence of the small-compound (**Figure 3G** and **Table 2**).

## EMD 57033-mediated refolding of myosin

Equilibrium binding of ligands leads to a shift of the midpoint of the thermal transition $T_m$, which corresponds to the temperature at which half of all protein molecules are in the native state and the remaining half are in the denatured state. Typically, the extent of the observed increase in protein thermal stability is proportional to the concentration and affinity of the added ligand. The $T_m$ measured for the *Dd* myosin-2 motor domain corresponds to 45.6 ± 0.2°C in the absence of nucleotide and 52.7 ± 0.2°C in the presence of 400 µM Mg$^{2+}$-ADP•BeF$_3$ (**Ponomarev et al., 2000**). In the case of the β-cardiac myosin motor domain, we measured $T_m$ values of 45.8 ± 2.2°C (nucleotide-free) and 52.0 ± 0.8°C (+100 µM Mg$^{2+}$-ATP) in low salt buffer and 46.5 ± 1.1°C (nucleotide-free) and 54.1 ± 0.9°C (+100 µM Mg$^{2+}$-ATP) in high salt buffer, respectively. The addition of a saturating concentration of EMD 57033 to nucleotide-free β-cardiac myosin motor domain shifts the $T_m$ to 53.6 ± 1.9°C (low salt) (**Figure 4—figure supplement 1**) and for the complex with F-actin from 55.5 ± 3°C to 66.4 ± 3°C (**Figure 4—figure supplement 2**). However, the EMD 57033-mediated activation and conversion of inactive to catalytically competent myosin (**Figure 1C,D**) and the greatly improved preservation of enzymatic activity during storage

**Table 2.** EMD 57033-mediated changes in the kinetic behavior of **β**-cardiac myosin

|  | Control | 25 µM EMD 57033 | ~Change (−fold) |
|---|---|---|---|
| $K_{M(Actin)}$ | 36. 8 ± 2.67 µM | 3. 6 ± 0.8 µM | 10 |
| $k_{cat}$ | 0. 12 ± 0.02 s$^{-1}$ | 0.19 ± 0.02 s$^{-1}$ | 1.6 |
| $k_{cat}/K_{M(Actin)}$* | 0.00326 µM$^{-1}$ s$^{-1}$ | 0.0528 µM$^{v1}$ s$^{-1}$ | 16 |
| $k_{cat}/K_{M(Actin)}$† | 0.00193 µM$^{-1}$ s$^{-1}$ | 0.0304 µM$^{-1}$ s$^{-1}$ | 15.8 |
| $K_1 k_{+2}$ | 0.19 ± 0.02 µM$^{-1}$s$^{-1}$ | 0.65 ± 0.04 µM$^{-1}$s$^{-1}$ | 3.4 |
| $k_{+2}$ | 46 ± 3 s$^{-1}$ | 174 ± 7 s$^{-1}$ | 3.8 |
| $k_{+3} + k_{-3}$ | 1.9 ± 0.2 s$^{-1}$ | 24.5 ± 1.9 s$^{-1}$ | 12.9 |
| $k_{+4}$‡ | 0.09 ± 0.01 s$^{-1}$ | 0.15 ± 0.01 s$^{-1}$ | 1.7 |
| $k_{-D}$ | 0.15 ± 0.03 s$^{-1}$ | 0.16 ± 0.04 s$^{-1}$ | n.a. |
| $k_{-AD}$ | 25.5 ± 3 s$^{-1}$ | 26.9 ± 4 s$^{-1}$ | n.a. |
| $E_a$ | 47 ± 4 kJ mol$^{-1}$ | 31 ± 3 kJ mol$^{-1}$ | 1.5 |

*The apparent second order rate constant for actin binding ($k_{cat}/K_{M(Actin)}$) was obtained from the calculated ratio of both values.
†$k_{cat}/K_{M(Actin)}$ was obtained from the initial slope of the steady-state ATPase activity vs the F-actin plot.
‡Measured at F-Actin concentration of 25 µM and 25 µM ATP.

(*Figure 4A*) indicate that EMD 57033 is more potent in preventing the precipitation and irreversible denaturation of the protein than suggested by the shift in $T_m$ alone.

To elucidate the mode of action of EMD 57033, we submitted myosin constructs in the absence and presence of the compound to heat stress and followed the resulting changes in solubility and functional competence. The fraction of myosin that is able to bind mantATP drops to below 32% when the temperature is raised to 49°C for more than 5 min. Incubation for 10 min at 51°C removes the myosin completely and irreversibly from the soluble fraction. Preincubation of the myosin with 25 µM EMD 57033 before exposure to heat-stress prevents precipitation (*Figure 4B*, insert). Moreover, based on the amplitude of the rise in mantATP fluorescence, the competence of β-cardiac myosin to bind nucleotide is preserved to more than 91% after 5-min incubation at 49°C and to approximately 82% after 10-min incubation at 49°C (*Figure 4B*). The apparent large reduction in the rate of mantATP binding and the extended plateau phase suggest that the observed transients report a rate-limiting step in the refolding of the β-cardiac myosin motor rather than the rate of nucleotide binding (see *Figure 1B* for comparison). This view is supported by experiments that monitor the recovery of basal activity over a 2 hr period (*Figure 4C*). To obtain more detailed information about the kinetics of the EMD 57033-mediated refolding reaction, we performed three sets of experiments recording the time and concentration dependence of EMD 57033-mediated increases in the fraction of myosin molecules that are able to undergo nucleotide induced conformational changes, hydrolyze ATP, and produce movement in the in vitro motility assay.

Changes in intrinsic protein fluorescence are tightly linked to ATP binding and ATP-induced conformational changes in the myosin motor domain (*Batra and Manstein, 1999*). Therefore, the progress of the refolding reaction can be estimated by incubating heat-inactivated β-cardiac myosin with EMD 57033 and recording the change in intrinsic protein fluorescence that follows the addition of substoichiometric amounts of ATP (*Figure 4D*) and excess ATP (*Figure 4E*) at predefined time points over the next two hours. The dependence of the $k_{obs}$ values on the incubation time was best fit by a hyperbola (*Figure 4E*). Refolding and reactivation of ATPase activity and motility were measured in the presence of 10–100 µM EMD 57033. In each experimental setting the data obtained for a specific EMD 57033 concentration were best fit by hyperbolae (*Figure 4F,G*). The recovery of motor activity upon addition of EMD 57033 to flow cells is shown in *Video 1* and *Video 2*. The $k_{obs}$ values derived from the hyperbolae determined for EMD 57033-mediated changes in ATPase activity and motility (*Figure 4F,G*) and from changes in intrinsic protein fluorescence (data not shown) indicate a linear dependence between the progress of the refolding reaction and the concentration of EMD 57033. The slope defines a value of $0.02 \times 10^{-3}$ µM$^{-1}$ s$^{-1}$ for the second order rate constant for the EMD 57033-mediated refolding reaction (*Figure 4H*).

## EMD 57033-mediated cellular effects

Our studies provide evidence for a pro-hypertrophic effect of EMD 57033 (*Figure 5A,B*). After 1 day exposure to 10 µM EMD 57033 in serum-free growth medium, neonatal rat cardiomyocytes (NRCM) displayed a 1.8- and 1.4-fold increase in cell size without significant changes in α-cardiac MHC transcripts and a reduction in β-cardiac MHC transcripts without significant changes in total cardiac MHC protein content compared to controls (*Figure 5C–F*). Atrial natriuretic peptide (ANP) has been described as a marker that rapidly responds to increased cardiac strain and stress induced by hyperthermia (*Aggeli et al., 2002*; *Chen et al., 2012*). ANP expression was not significantly changed in NRCM treated with 10 µM EMD 57033 (*Figure 5G*).

Hyperthermic stress for 24 hr did not significantly alter cell size or MHC total protein content in NRCM but reduced transcript levels of both α- and β-cardiac MHC (*Figure 5C–F*). The reduction of β-cardiac MHC remained more pronounced in the EMD-treated NRCM without affecting MHC protein content (*Figure 5*). Hyperthermic stress-induced ANP expression in controls, while the presence of 10 µM EMD 57033 fully suppressed the expression of ANP during hyperthermic stress (*Figure 5G*).

## Discussion

Our results show that EMD 57033 is a member of a new class of pharmacological chaperones that stabilize, enhance the activity, and correct stress-induced misfolding of their target protein. EMD 57033 binds to an allosteric pocket in the myosin motor domain. The site appears to be close to the site where cardiac myosin activator omecamtiv mecarbil is predicted to bind (*Malik et al., 2011*). The greater isoform specificity of omecamtiv mecarbil and preliminary results with other compounds suggest that it is feasible to identify compounds with improved properties for the treatment of aberrant myosin motor activity that are derived from thiadiazinone derivatives and unrelated scaffolds. The efficacy of the resulting drugs

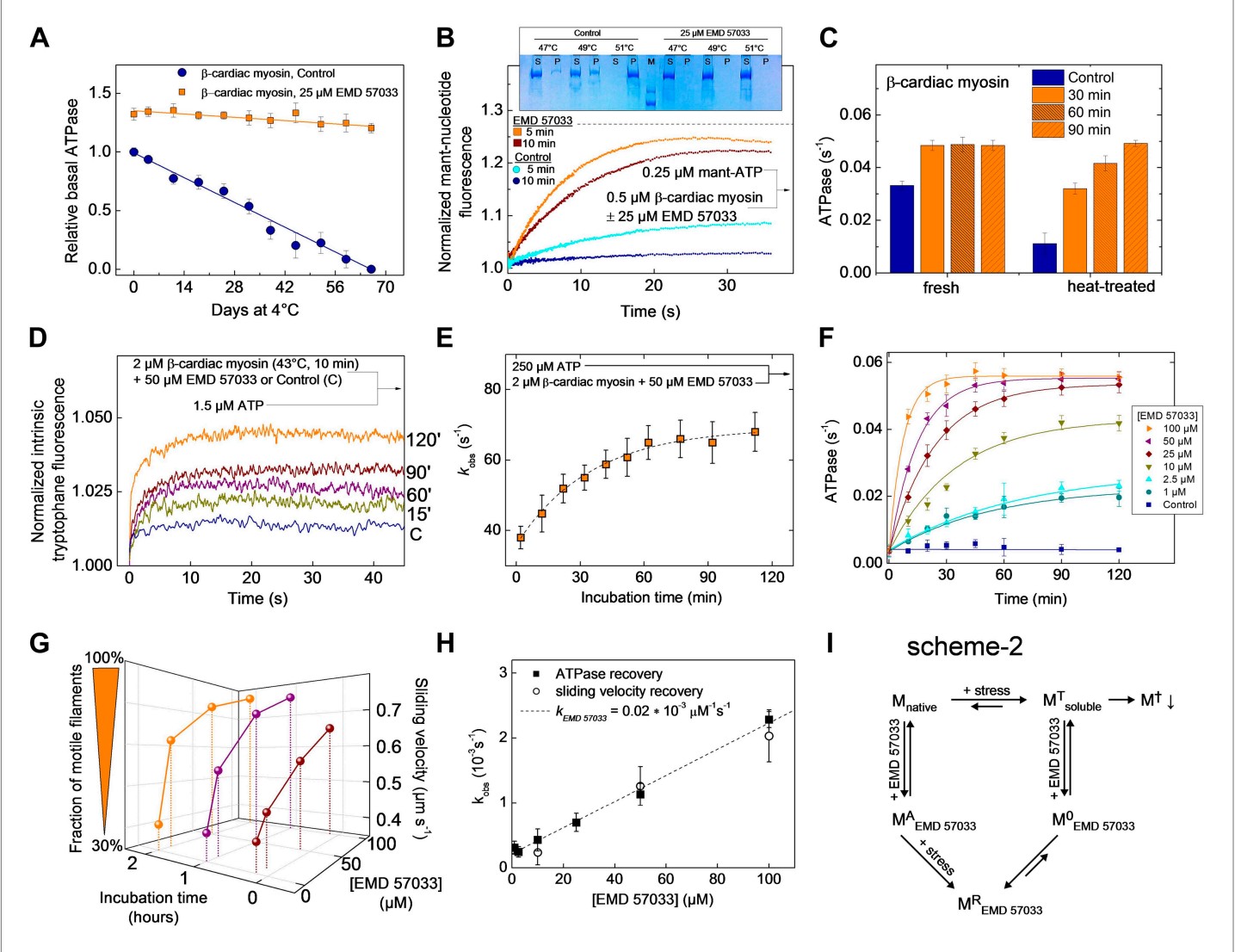

**Figure 4**. EMD 57033 acts as a pharmacological chaperone. (**A**) The presence of 25 µM EMD 57033 extends the shelf life of *Hs* β-cardiac myosin at 4°C. Errors indicate SD (n = 3). (**B**) EMD 57033 renders β-cardiac myosin more heat-stable. The SDS-PAGE gel shows supernatants (S) and pellets (P) after 10-min incubation of β-cardiac myosin at the indicated temperatures. In the presence of EMD 57033 β-cardiac myosin remained in the supernatant over the entire temperature range tested. Incubation for 10 min at 49°C leads to the almost complete loss of catalytic activity. The ability to bind mantATP is gradually recovered following the addition of 25 µM EMD 57033. The dotted line represents the fluorescence amplitude for mantATP binding to native β-cardiac myosin. (**C**) Rescue and activation of heat-treated β-cardiac myosin basal ATPase activity by EMD 57033. The orange columns indicate changes observed in the presence of EMD 57033. Errors indicate SD (n = 3). (**D**) Time-dependent recovery of the capacity to bind nucleotide monitored by ATP-induced changes in intrinsic tryptophan fluorescence. Refolding was initiated by the addition of 50 µM EMD 57033. Transients obtained after rapid mixing with ATP were measured at the indicated times following the addition of EMD 57033. (**E**) The observed rate constants for ATP binding to heat-treated β-cardiac display a hyperbolic dependence upon the incubation with EMD 57033. Errors indicate s.d. (**F**) Effect of EMD 57033 concentration on the time-dependent recovery of heat-treated β-cardiac myosin ATPase activity. Errors indicate SD (n = 3). (**G**) Time and EMD 57033 concentration dependence of the recovery of β-cardiac myosin motor activity. (**H**) Determination of the second order rate constant for the EMD 57033-mediated refolding reaction. The observed rate constants for the recovery of ATPase activity and motility were extracted from the data shown in ***Figure 4F,G***. The slope of a linear fit to the data defines $k_{rescue}$ as $0.02 \times 10^{-3}$ µM$^{-1}$s$^{-1}$. The y-intercept gives a first order dissociation rate constant for EMD 57033 of $0.23 \times 10^{-3}$ s$^{-1}$ (**I**) Proposed model for the mode of action of EMD 57033. [T] Conformationally trapped soluble aggregates, nucleotide binding incompetent; [R] Rescued, hydrolysis and motility competent; [0] Compromised, hydrolysis incompetent, nucleotide binding competent; [A] Activated; [†] Insoluble protein aggregates (irreversible); [↓] Precipitation.

*Figure 4. Continued on next page*

*Figure 4. Continued*

The following figure supplements are available for figure 4:

**Figure supplement 1**. Melting curves of the β-cardiac myosin motor domain as determined by circular dichroism spectroscopy at 222 nm (c = 0.3 mg/ml).

**Figure supplement 2**. Melting curves of the complex formed by the β-cardiac myosin motor domain with F-actin followed by the change in light-scattering signal.

may be further enhanced by combining them with drugs that increase the abundance and activity of cellular chaperones (*Willis and Patterson, 2013*). The actin-affinity of cardiac β-myosin and the coupling efficiency between motor domain and actin filament are increased in the presence of saturating concentrations of EMD 57033. The selective shortening of the lifetime of weakly bound states and the associated increase in the fraction of strongly bound motor domains explain the leftwards shift of the force–pCa relation and the reduced oxygen cost of contractility observed in previous studies (*Solaro et al., 1993*; *Kraft and Brenner, 1997*; *Senzaki et al., 2000*). The expected concomitant increase in maximal isometric force production is confirmed by the results obtained using the frictional load assay.

In addition to promoting the formation of a super-active $M^A_{EMD\,57033}$ state, binding of EMD 57033 increases the thermal stability of the cardiac β-myosin motor domain. The observed shifts in melting temperature are significant but do not exceed levels observed with high-affinity binders such as enzyme substrates, substrate analogues, and competitive and allosteric inhibitors. However, in the presence of EMD 57033 the formation of insoluble $M^\dagger$ aggregates is reduced to a much greater extent than observed for competitive inhibitors and other stabilizing agents. This more potent effect of EMD 57033 on the retention of enzymatic activity and maintenance of near structural integrity appears to be mediated by its interaction with stress-induced misfolded but soluble $M^T_{soluble}$ intermediates as described in scheme 2 (*Figure 4I*). EMD 57033 binding to $M^T_{soluble}$ induces the transition of the enzymatically inactive myosin to a nucleotide binding-competent state $M^0_{EMD\,57033}$, followed by the transition to a super-active $M^R_{EMD\,57033}$ state that displays similar enzymatic activity as the EMD 57033-activated native enzyme. The need for separate $M^A_{EMD\,57033}$ and $M^R_{EMD\,57033}$ states arises from our observation that only myosin that has not been exposed to any kind of stress can be returned to the $M_{native}$ state, when the compound is washed out.

Many genetically transmitted diseases lead to mutant proteins with reduced stability. The resulting proteotoxicity is frequently of equal or greater importance for the clinical expression of the associated disease states as the direct, mutation-induced loss in enzymatic activity. Misfolded proteins including mutation-induced misfolded myosins are known to play a central role in the pathophysiology of neurodegenerative diseases. The role of misfolded proteins in the development of numerous other disorders such as type-2 diabetes and heart failure has become evident in recent years (*Willis and Patterson, 2013*). Moreover, cardiac dysfunction is a frequent complication associated with elevated body temperature due to fever or hyperthermia (*Marijon et al., 2012*). It is well known that pathophysiologic alterations of the heart are associated with a decrease in α-MHC and an increase in β-MHC expression (*Mercadier et al., 1983*). This is normally associated with an increase in ANP (*Aggeli et al., 2002*). In fact, ANP levels in the myocardium are greatly augmented in patients with congestive heart failure and animal models of ventricular hypertrophy or cardiomyopathy (*Ogawa et al., 1995*). Such changes are indicative of re-expression of a fetal gene expression program, as it is induced by the α-adrenergic agonist phenylephrine (*Hilfiker-Kleiner et al., 2006*). Physiological stress such as exercise induces a shift towards α-cardiac MHC (*Allen et al., 2001*; *Baldwin and Haddad, 2001*). EMD 57033 induces hypertrophic growth under normothermic conditions that involves a reduction in β-MHC without affecting α-cardiac MHC and thereby changes in

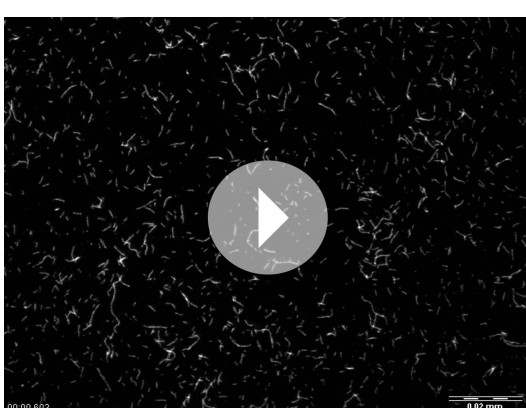

**Video 1**. Video showing fluorescently labeled actin filaments attached to a lawn of catalytically inactive β-cardiac myosin. (MT state, *Figure 4I*).

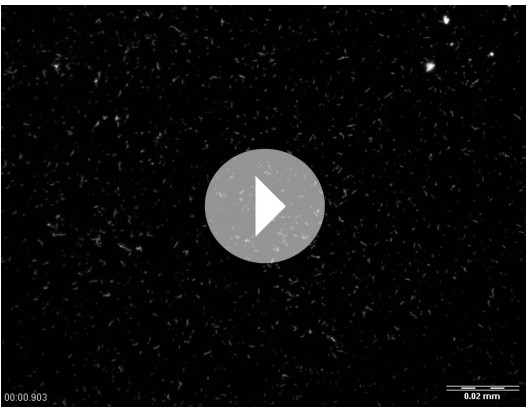

**Video 2**. Reactivation of β-cardiac myosin by the addition of EMD 57033. Video showing motile actin filaments on the same lawn of β-cardiac myosin three hours after the addition of 10 µM EMD 57033 to the flow cell (M$^R$ state, **Figure 4I**).

the transcriptional regulation that are clearly distinct from phenylephrine-mediated cardiomyocyte hypertrophy (*Hilfiker-Kleiner et al., 2006*). Therefore, EMD 57033 appears to promote a more physiological type of hypertrophy. This notion is further supported by the observation that ANP expression is not induced in the presence of EMD 57033. In addition, the observed reduction of ANP expression in heat-treated cardiomyocytes suggests that EMD 57033-mediates a reduction in cellular stress responses by stabilizing and refolding denatured MHC proteins.

Our results provide evidence for the refolding effect exerted by a small drug-like molecule on a complex protein such as myosin. It remains to be shown whether the refolding activity is shared by other compounds binding to the same region of myosin and to what extent it can be separated from the activation of chemomechanical activity. Small chemical compounds with similar activity profile but increased specificity for their target protein have the potential to improve the treatment of protein misfolding diseases, myopathies, and heart failure.

The functionalization of biohybrid devices that exploit actomyosin-based cargo transport for molecular diagnostics and other nanotechnological applications (*Amrute-Nayak et al., 2010*; *Lard et al., 2013*; *Persson et al., 2013*) is another area that will certainly benefit from the use of EMD 57033 and the development of compounds that stabilize their target proteins against stress-induced denaturation, precipitation, and proteolytic degradation. The effective and simple long-term storage of chip-based nanobiosensors, where actomyosin-driven transport substitutes microfluidics and forms the basis for novel detection schemes, is a precondition for the commercial viability of such devices. The addition of EMD 57033 extends the shelf life of protein-functionalized surfaces from a couple of hours to several months, making the widespread and routine use of biohybrid devices feasible.

## Materials and methods

### Transient kinetics

Stopped-flow experiments were performed using PiStar (Applied Photophysics, Leatherhead, UK) and Hi-Tech SF61 (TgK Scientific Limited, Bradford on Avon, UK) spectrophotometers at 20°C as described previously (*Taft et al., 2008*). Mant-nucleotides (Jena Bioscience, Germany) were excited at 365 nm and detected after passing through a KV 389 nm cut-off filter. Tryptophane fluorescence was excited at 297 nm and detected using a WG 320 nm cut-off filter. Long-time experiments were carried out using a shutter to avoid photo-bleaching. To account for possible absorption or fluorescence effects, control measurements were carried out in the absence of EMD 57033, as well as in the presence of the non-binding (−)-enantiomer EMD 57439. Phosphate release was measured using MDCC-PBP (N-[2-(1-maleimidyl)ethyl]-7-(diethylamino)coumarin-3-carboxamide fused to phosphate binding protein; obtained from Life Technologies, Carlsbad, CA) as described previously (*Brune et al., 1994*). Coumarin fluorescence was excited at 430 nm and detected using 455 nm cut-off-filter.

### Microscale thermophoresis

We used recombinant myosin constructs fused to yellow fluorescent protein (*Dd* myosin-2, *Dd* myosin-1B, *Dd* myosin-1C, *Dd* myosin-1D, *Dd* myosin-1E) or myosins chemically labeled with an amine reactive RED-NHS dye (*Dd* myosin-2, β-cardiac myosin). Experiments were performed using hydrophobic capillaries in a NanoTemper Monolith NT.115 instrument according to Duhr et al. (*Duhr and Braun, 2006*; *Wienken et al., 2010*).

### Protein preparation

Rabbit fast skeletal muscle heavy meromyosin (HMM) was prepared as described by *Margossian and Lowey (1982)*. Porcine β-cardiac myosin was prepared essentially as described by *Pant et al. (2009)*

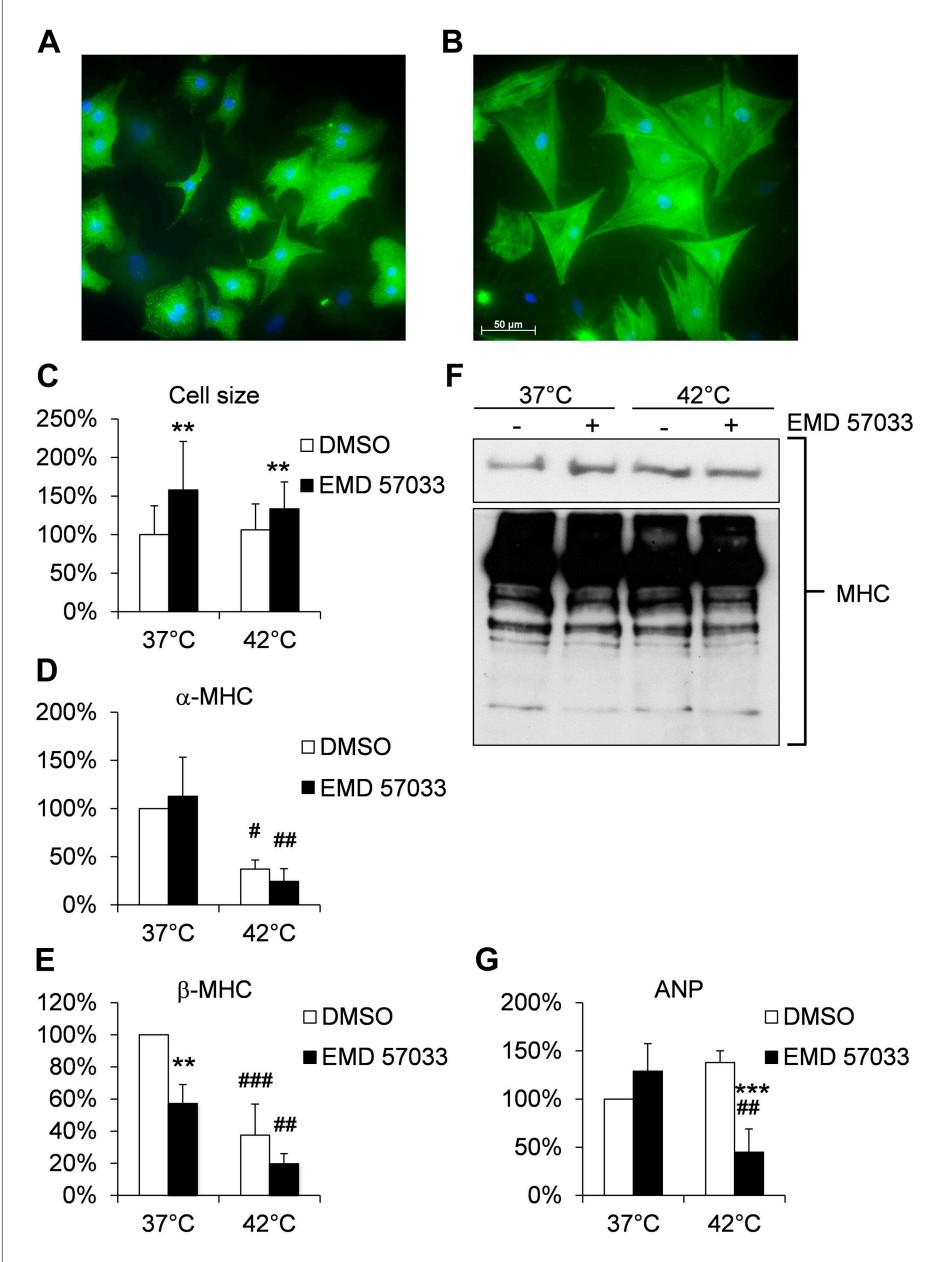

**Figure 5**. Effects of EMD 57033 on neonatal rat cardiomyocytes (NRCM) in vitro. (**A**) Cell size and organization of the sarcomere in representative NRCM (DMSO control) at 37°C. (**B**) Representative NRCM after 24 hr incubation with EMD 57033. Cardiomyocytes were immunostained for α-actinin (green) and DAPI was used for nuclear staining. (**C**) Bar graph depicting cumulative measurements of cell size of NRCM after 24 hr incubation with EMD 57033 or DMSO at 37°C or 42°C. NRCM display a comparable increase in cell size in response to EMD 57033 stimulation independent of the incubation temperature. (**D**) Bar graph showing expression of α-cardiac MHC mRNA normalized to b2m in NRCM after 24 hr treatment with EMD 57033 or DMSO. EMD 57033 does not alter α-cardiac MHC mRNA expression compared to DMSO controls. Incubation at 42°C significantly reduces expression of α-cardiac MHC mRNA in NRCM compared to cells that were incubated at 37°C. The decrease in α-cardiac MHC mRNA upon induction of hyperthermia was not affected by EMD 57033 treatment. Errors indicate SD (n = 3). (**E**) Bar graph summarizing mRNA expression of β-cardiac MHC normalized to b2m in EMD 57033 or DMSO-treated NRCM after 24-hr incubation at 37°C or 42°C. Treatment with EMD 57033 significantly reduces β-cardiac MHC expression at 37°C compared to DMSO controls. Incubation at 42°C for 24 hr results in downregulation of β-cardiac MHC mRNA. Downregulation is significantly more pronounced in EMD 57033-treated cells. Errors indicate SD (n = 3). (**F**) Representative immunoblot depicting expression of total cardiac MHC protein content in NRCM treated with
*Figure 5. Continued on next page*

*Figure 5. Continued*

EMD 57033 or DMSO. EMD 57033 treatment does not significantly alter MHC protein levels at 37°C or 42°C and MHC protein content is comparable in NRCM incubated at 37°C or 42°C. (**G**) Bar graph depicting mRNA expression of ANP normalized to b2m in NRCM treated with EMD 57033 or DMSO for 24 hr at 37°C or 42°C. EMD 57033 treatment does not affect expression of ANP mRNA at 37°C. Incubation of NRCM at 42°C induces mRNA expression of ANP in DMSO-treated control cells, which is completely suppressed by treatment with EMD 57033. Errors indicate SD (n = 3). NRCM were treated with EMD 57033 (10 µM) or the solvent DMSO (1 µl/ ml) alone for 24 hr in serum-free conditions. Significances are coded as follows: *p<0.05, **p<0.01 und ***p<0.001 for the DMSO control vs EMD 57033 at a specific temperature; #p<0.05, ##p<0.01 und ###p<0.001 for a specific group at different temperatures.

and *Jacques et al. (2008)* and further purified by gel filtration. S1 and HMM fragments were produced by proteolytic digestion of the full-length proteins with 0.05 mg/ml papain (10 min, 25°C) for S1 and with 0.1 mg/ml TLCK-treated α-chymotrypsin (15 min, 25°C) for HMM. Reactions were stopped with specific protease inhibitors for papain (10 µM E−64) and chymotrypsin (10 µM chymostatin, 1 mM benzamidine). Proteolytic fragments were further purified by actin-coprecipitation. Pellets were washed twice and myosin was extracted from the pellets using ATP containing buffer. Gel filtration was used to further increase purity of myosin S1 and HMM fragments. His-tagged motor domain constructs of *Dd* myosin-2, *Dd* myosin-1B, *Dd* myosin-1C, *Dd* myosin-1D, *Dd* myosin-1E, *Dd* myosin-5b fused to two *Dd* α-actinin repeats substituting the light-chain binding domains (*Anson et al., 1996*) were expressed in *Dd* AX3-ORF + cells and purified by Ni$^{2+}$-chelate affinity chromatography (*Manstein and Hunt, 1995*). F-actin was prepared by the method of *Lehrer and Kerwar (1972)*.

## Determination of protein concentration and the relative amount of active myosin

We used the molar absorption coefficients at 280 nm to determine the concentration of the myosin constructs in solution. Fast mixing active-site titrations were performed to follow the fluorescence increase of mantATP binding to myosin. We determined the minimal mantATP concentration necessary to fully saturate the binding amplitude. This mantATP concentration defines the actual concentration of active myosin heads in the respective protein preparation.

## Steady-state kinetics

Basal and actin-activated Mg$^{2+}$-ATPase activities were measured using the NADH-coupled assay described previously (*Furch et al., 1998*) adapted for higher throughput screens for a temperature controlled plate reader (Multiscan FC, Thermo Scientific, Waltham, MA) at 25°C. EMD 57033 was added to the reaction mixture in the absence of nucleotide and incubated before the reaction was started by the addition of ATP. Control measurements were carried out in the absence of drug, as well as in the presence of according concentrations of the non-binding (−)-enantiomer EMD 57439 to account for possible absorption effects. Each reaction mixture including the controls contained 2.5% DMSO that was used as solvent for EMD 57033. Data points are mean values from 3 to 5 independent experiments. Error bars indicate standard deviations.

## In vitro motility assay

Actin-sliding motility was performed at 20°C using an Olympus IX81 (Olympus, Hamburg, Germany) inverted fluorescence microscope as described previously (*Fujita-Becker et al., 2005*). Average sliding velocities were determined from the Gaussian distribution of automatically tracked actin filaments using DiaTrack 3.01 (Semasopht, Chavannes, Switzerland) and Origin 7.0 (OriginLab Corporation, Northampton, MA). Errors indicate standard deviations.

## Melting temperature determination

Circular dichroism and differential static light scattering experiments were performed to determine $T_m$.

### Circular dichroism

Protein solutions were diluted to a concentration of 0.3 mg/ml in a buffer containing 5 mM KP$_i$ pH 7.5, 15 mM KCl. Denaturation of β-cardiac myosin was measured by monitoring the temperature-dependent changes of ellipticity at 222 nm in a temperature controlled π*-180 Spectrometer equipped with a circular dichroism unit (Applied Photophysics, Leatherhead, UK). Measurements were performed

in a 0.3-cm path length Quartz cuvette with a temperature gradient of 1°C min$^{-1}$. The transition midpoint of the unfolding reaction was calculated by fitting a Boltzmann function to the experimental data. Prior to measurements, the storage buffer of β-cardiac myosin that contains DTT and sucrose were exchanged using PD MiniTrap G-25 (GE Healthcare, Little Chalfont, UK) size exclusion columns equilibrated with CD Buffer (5 mM KP$_i$ pH 7.5, 15 mM KCl).

## Differential Static Light Scattering

Protein stability was assayed following protein aggregation associated with heat denaturation (*Senisterra et al., 2006*). Protein solutions were diluted to a concentration of 0.2 mg/ml in a buffer containing 5 mM KP$_i$ pH 7.5, 25 mM KCl and heated in steps of 0.6°C. Static light scattering was measured at each temperature step and the resulting transition curve was fitted by a Boltzmann function. The transition midpoint gives a measure for the aggregation temperature induced by protein denaturation.

## Molecular Modeling

Homology models of the *Hs* β-cardiac myosin-2 motor domain (residues 1–800) were built using Modeller (*Sali and Blundell, 1993*) and the X-ray crystal structure of *Gg* myosin-2 Subfragment-1 (PDB code: 2MYS) as template. Prior to molecular docking, the protein models were subjected to geometry optimization using MacroModel (MacroModel, version 9.9; Schrödinger, LLC, New York, NY, 2011) and the OPLS2005 force field. Blind and local docking of EMD 57033 was carried out using Autodock4 (*Morris et al., 1998*), employing the Lamarckian Genetic Algorithm. The ligand was prepared and energy-minimized using MacroModel (MacroModel, version 9.9; Schrödinger, LLC, New York, NY, 2011) and the OPLS2005 force field, as well as AutodockTools (*Sanner, 1999*; *Morris et al., 2009*). During flexible docking, the side chains of Arg29 and Lys34 were allowed full conformational flexibility. In addition, local docking was performed using models that are based on the X-ray structure of *Hs* β-cardiac myosin-2 experimentally solved by the laboratory of I Rayment (PDB code: 4DB1).

## Isolation, culture and treatment of NRCM

Cell culture media, fetal bovine serum (FBS), and horse serum (HS) were purchased from Biochrome (Berlin, Germany); all other chemicals were from Sigma-Aldrich (St Louis, MO). Isolation of neonatal rat cardiomyocytes (NRCM) was performed by enzymatic digestion of neonatal rat hearts as described previously (*Hilfiker-Kleiner et al., 2004*). NRCM were cultured in plating medium containing 5% FBS and 10% HS for 24 hr followed by serum-starvation for 48 hr. NRCM were cultured in serum-free medium containing either EMD (10 μM) or the corresponding volume of the solvent DMSO (1 μl/ ml) for additional 24 hr at 37°C or 42°C with 5% $CO_2$.

## Immunofluorescence

For indirect immunofluorescence NRCM were fixed with 95% ethanol. Monoclonal antibodies against sarcomeric α-actinin (1:50; Sigma-Aldrich) or sarcomeric Myosin heavy chain (1:50; Santa Cruz Biotechnology, Dallas, TX) were used. Fluor-488 goat anti-mouse IgG (1:250; Jackson ImmunoResearch, West Grove, PA) or Alexa Fluor 639 goat anti-rabbit (1:250; Jackson ImmunoResearch) was employed as secondary antibody. Immunofluorescence was detected by the use of the AxioVison Rel 4.1 package (Carl Zeiss GmbH).

## Realtime PCR

Realtime measurement of PCR amplification was performed using the Stratagene MX4000 multiplex QPCR System with the SYBR green dye method (Brilliant SYBR Green Mastermix-Kit, Stratagene, La Jolla, CA) as described (*Hilfiker-Kleiner et al., 2010*; *Hoch et al., 2011*). Total RNA was extracted from NRCM using Trizol reagent according to the manufactures protocol (Invitrogen, Carlsbad, CA). Reverse transcriptase (RT)-PCR was performed of ANP and b2m using 2 μg of total RNA. The primer sequences and PCR conditions are described in *Table 3*. PCR products were size-fractionated by 2% agarose gel electrophoresis.

## Immunoblotting

Protein expression in NRCM was determined by standard immunoblotting techniques under denaturing conditions, as described previously (*Hilfiker-Kleiner et al., 2004*). A monoclonal primary antibody against sarcomeric MHC (1:1.000; Abcam, Cambridge, UK) was used and signal detection was achieved by the usage of a horseradish peroxydase-conjugated secondary antibody (GE) and ECL-detection. PonceauS staining was carried out and used as a loading control.

**Table 3.** Primer sequences and PCR conditions used for realtime PCR

| Primer | Primer-Sequence | Annealing temperature (°C) |
|---|---|---|
| r-ANP forward | 5'-GCCGGTAGAAGATGAGGTCA-3' | 60 |
| r-ANP reverse | 5'-GGGCTCCAATCCTGTCAATC-3' | 60 |
| r-aMHC forward | 5'-GGAAGAGCGAGCGGCGCATCAAGG-3' | 55 |
| r-aMHC reverse | 5'-GTCTGCTGGAGAGGTTATTCTCG-3' | 55 |
| r-bMHC forward | 5'-CAAGTTCCGCAAGGTGC-3' | 55 |
| r-bMHC reverse | 5'-AAATTGCTTTATTGTGTTTCT-3' | 55 |
| r-b2m forward | 5'-CATGGCTCGCTCGGTGACC-3' | 60 |
| r-b2m reverse | 5'-AATGTGAGGCGGGTGGAACTG-3' | 60 |

# Acknowledgements

We thank B Brenner, C Gregor, N Hundt, P Kay-Fedorov, and N Steinke for help and discussions; D Wienke and N Beier for discussions; Merck KGaA, Darmstadt, Germany for providing EMD 57033 and EMD 57439. MBR participated in the PhD Program 'Molecular Medicine' of Hannover Biomedical Research School.

# Additional information

## Funding

| Funder | Grant reference number | Author |
|---|---|---|
| European Commission | 228971 | Dietmar J Manstein |
| Deutsche Forschungsgemeinschaft | EXC62/2 | Denise Hilfiker-Kleiner, Dietmar J Manstein |

The funders had no role in study design, data collection and interpretation, or the decision to submit the work for publication.

## Author contributions

MBR, MHT, Acquisition of biochemical and biophysical data, Analysis and interpretation of data, Drafting or revising the article; BS, Acquisition of cell-based data, Analysis and interpretation of data; DH-K, Conception and design of cell-based assays, Acquisition of data, Analysis and interpretation of data, Drafting or revising the article; MP, Conceived and performed molecular modelling and docking, Analysis and interpretation of data, Drafting or revising the article; DJM, Supervised the project, Conception and design, Analysis and interpretation of data, Drafting or revising the article, Contributed unpublished essential data or reagents

# Additional files

## Major dataset

The following previously published datasets were used:

| Author(s) | Year | Dataset title | Dataset ID and/or URL | Database, license, and accessibility information |
|---|---|---|---|---|
| Rayment I, Rypniewski WR, Schmidt-Base K, Smith R, Tomchick DR, Benning MM, Winkelmann DA, Wesenberg G, Holden HM | 1993 | Three-dimensional structure of myosin subfragment-1: a molecular motor | 2MYS; http://www.rcsb.org/pdb/explore/explore.do?structureId=2mys | Publicly available at RCSB Protein Data Bank (http://www.rcsb.org). |

| Klenchin VA, Deacon JC, Combs AC, Leinwand LA, Rayment I | Cardiac human myosin S1dC, beta isoform complexed with Mn-AMPPNP | 4DB1; http://www.rcsb. org/pdb/explore/explore. do?structureId=4db1 | Publicly available at RCSB Protein Data Bank (http://www.rcsb.org). |

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
