## [Decision Letter]

Thank you for sending your work entitled “Small molecule-mediated Refolding and Activation of Myosin Motor Function” for consideration at *eLife*. Your article has been favorably evaluated by a Senior editor and 2 reviewers, one of whom is a member of our Board of Reviewing Editors.

The Reviewing editor and the other reviewer discussed their comments before we reached this decision, and the Reviewing editor has assembled the following comments to help you prepare a revised submission.

The manuscript entitled “Small Molecule-mediated Refolding and Activation of Myosin Motor Function” by Radke et al. is a very interesting paper worthy of serious consideration for publication in *eLife*. While one can quibble with a few of the experiments or their description in this paper, collectively the data provide compelling evidence for the hypothesis that EMD 57033 is a pharmacologic chaperone that stabilizes, enhances the activity of, and corrects stress-associated misfolding of a subset of members of the myosin family comprising a SH3-like Beta barrel domain. I am not aware of another pharmacologic chaperone that can mediate refolding and gain of enhanced function from a denatured protein like EMD 57033 can with myosin.

1) Generally the experiments are very well executed. What is not clear from the text or the associated Figure 1 is whether EMD 57033 was added after the four typical preparations of beta-cardiac myosin (affording 32–90% activity) or whether it was present during the preparations leading to 87–98% activity. The implication of the text is that the 32-90% activity was converted to 87–98% activity by adding EMD 57033 after misfolding, i.e., that EMD 57033 mediated activation and conversion of inactive to catalytically hyperactive myosin. Please clarify these points in the text.

2) What is missing in the Abstract is a concluding sentence focusing on what the novel pharmacologic application would be for this unique pharmacologic chaperone-what can it be used for and why it is important. My impression is that it could be used to treat heart disease. If so, say so, or otherwise explain its importance.

3) The Discussion needs a further explanation of the potential medical uses of this very interesting pharmacologic chaperone and a further explanation of how the authors anticipate the application.

4) The claim that the authors have identified the site that EMD 57033 binds to (the pocket between the lever arm and small N-terminal, SH3-like barrel subdomain) is based mainly on the lack of activation of sliding velocity by EMD 57033 when the SH3 domain is deleted from the protein and on molecular docking studies. In the absence of a co-crystal structure, I would prefer to see some point mutations of myosin made that eliminate key residues predicted to interact with EMD 57033 in the SH3-like domain or binding competition assays with omecamtiv since the authors mention the binding site for this molecule is nearby. While it would be valuable to see more evidence supporting the actual binding site, especially for the development of improved PCs or to accurately study the MOA of these molecules, I think it would be sufficient to state that the results support this site as the potential binding site, but further confirmation is needed.

---

## [Author Response]

*1) Generally the experiments are very well executed. What is not clear from the text or the associated*
Figure 1* is whether EMD 57033 was added after the four typical preparations of beta-cardiac myosin (affording 32*–*90% activity) or whether it was present during the preparations leading to 87*–*98% activity. The implication of the text is that the 32-90% activity was converted to 87*–*98% activity by adding EMD 57033 after misfolding, i.e., that EMD 57033 mediated activation and conversion of inactive to catalytically hyperactive myosin. Please clarify these points in the text*.

To clarify this point, we have added the following sentence to the main text: ”The fraction of active sites increased to 87–98% for the same enzyme preparations following the addition of EMD 57033.”

In addition, we changed the text of the figure legend: “The fraction of active protein increased to 87–100% following the addition of EMD 57033 (25 ?M for ?-cardiac myosin and 100 ?M for Dd myosin-2).”

*2) What is missing in the Abstract is a concluding sentence focusing on what the novel pharmacologic application would be for this unique pharmacologic chaperone-what can it be used for and why it is important. My impression is that it could be used to treat heart disease. If so, say so, or otherwise explain its importance*.

We added the following to the end of the Abstract: “Thus, EMD 57033 displays a much wider spectrum of activities than those previously associated with small, drug-like compounds. Allosteric effectors that mediate refolding and enhance enzymatic function have the potential to improve the treatment of heart failure, myopathies, and protein misfolding diseases.”

*3) The Discussion needs a further explanation of the potential medical uses of this very interesting pharmacologic chaperone and a further explanation of how the authors anticipate the application*.

We changed the concluding part of the Discussion:

“Our results provide evidence for the refolding effect exerted by a small drug-like molecule on a complex protein such as myosin. It remains to be shown whether the refolding activity is shared by other compounds binding to the same region of myosin and to what extent it can be separated from the activation of chemomechanical activity. Small chemical compounds with similar activity profile but increased specificity for their target protein have the potential to improve the treatment of protein misfolding diseases, myopathies and heart failure.

The functionalization of biohybrid devices that exploit actomyosin-based cargo transport for molecular diagnostics and other nanotechnological applications [Amrute-Nayak et al; Lard et al; Persson et al] is another area that will certainly benefit from the use of EMD 57033 and the development of compounds that stabilize their target proteins against stress-induced denaturation, precipitation, and proteolytic degradation. The effective and simple long-term storage of chip-based nanobiosensors, where actomyosin-driven transport substitutes microfluidics and forms the basis for novel detection schemes, is a precondition for the commercial viability of such devices. The addition of EMD 57033 extends the shelf life of protein-functionalized surfaces from a couple of hours to several months, making the widespread and routine use of biohybrid devices feasible.”

*4) The claim that the authors have identified the site that EMD 57033 binds to (the pocket between the lever arm and small N-terminal, SH3-like barrel subdomain) is based mainly on the lack of activation of sliding velocity by EMD 57033 when the SH3 domain is deleted from the protein and on molecular docking studies. In the absence of a co-crystal structure, I would prefer to see some point mutations of myosin made that eliminate key residues predicted to interact with EMD 57033 in the SH3-like domain or binding competition assays with omecamtiv since the authors mention the binding site for this molecule is nearby. While it would be valuable to see more evidence supporting the actual binding site, especially for the development of improved PCs or to accurately study the MOA of these molecules, I think it would be sufficient to state that the results support this site as the potential binding site, but further confirmation is needed*.

To clarify this point, we added the following sentence: “Further studies using point mutations that eliminate key residues predicted to interact with EMD 57033 are needed to confirm the binding site.”

To emphasize the preliminary nature of the identification of the EMD 57033 binding site, we also changed the text of the Abstract, the concluding part of the Introduction, and the Discussion.